# Occurrence of Melamine and Cyanuric Acid in Milk, Baby Food, and Protein Supplements Marketed in Croatia

**DOI:** 10.3390/foods14244292

**Published:** 2025-12-13

**Authors:** Bruno Čalopek, Nina Bilandžić, Ivana Varenina, Ines Varga, Božica Solomun Kolanović, Maja Đokić, Jelena Kaurinović, Renata Biba, Mario Cindrić, Marija Sedak

**Affiliations:** 1Laboratory for Residue Control, Department of Veterinary Public Health, Croatian Veterinary Institute, Savska cesta 143, 10000 Zagreb, Croatia; calopek@veinst.hr (B.Č.); kurtes@veinst.hr (I.V.); varga@veinst.hr (I.V.); solomun@veinst.hr (B.S.K.); dokic@veinst.hr (M.Đ.); kaurinovic@veinst.hr (J.K.); sedak@veinst.hr (M.S.); 2Laboratory for Bioanalytics, Ruđer Bošković Institute, Planinska 1, 10000 Zagreb, Croatia; mcindric@irb.hr (M.C.); renata.biba@irb.hr (R.B.)

**Keywords:** melamine, cyanuric acid, UHPLC-MS/MS, milk, baby food, protein supplements

## Abstract

The aim of this study is to investigate the occurrence of melamine and cyanuric acid in milk, baby food, and protein supplements collected in Croatia. A total of 56 samples were collected during 2022 and 2023 from retail stores in Zagreb, Croatia. Sample preparation involved acetonitrile extraction, followed by analysis using ultra-high-performance liquid chromatography coupled with tandem mass spectrometry (UHPLC-MS/MS). Cyanuric acid concentrations above the limit of quantification (LOQ) were found in five milk samples (33.3% detection frequency), with a range from 0.26 to 0.39 mg/kg and a mean concentration of 0.31 mg/kg. In protein supplements, melamine was detected above the LOQ in six samples (23% detection frequency), with a mean concentration of 0.30 mg/kg and concentrations ranging from 0.20 to 0.57 mg/kg. No concentrations above the LOQ were found in baby food samples. All detected values were below the EU maximum limits (2.5 mg/kg for general food and 1.0 mg/kg for baby food). The accuracy and reliability of the method were verified using certified reference material. This is the first study to confirm the presence of melamine and cyanuric acid in protein supplements and milk on the Croatian market. The detected levels do not indicate a potential health risk to consumers.

## 1. Introduction

With the increasing recognition of the nutritional benefits of milk, global consumption has risen significantly, resulting in heightened demand. However, a gap currently exists between supply and demand, as the surge in demand has not been matched by a corresponding increase in production capacity. Consequently, unethical practices, such as milk adulteration, have occasionally emerged in an attempt to enhance the quantity of product available in the market, often compromising its safety and nutritional value for the end consumer [1]. Milk adulteration is a significant global issue driven by several factors. The primary causes include the perishable nature of milk, the limited financial resources of certain consumers, and the absence of effective, accessible, and rapid analytical methods for real-time fraud detection [2]. In this context, the potential for manipulating the quality and composition of milk presents a considerable challenge to regulatory control and consumer protection systems.

Due to its high commercial value and widespread use in food production, milk powder is one of the most commonly adulterated food products in the world, second only to olive oil [2,3]. It serves as a fundamental raw material for the production of various food items, including infant formula, baked goods, confectionery products, and numerous industrial food preparations [2,3]. Economically motivated manipulations of milk and milk powder primarily diminish the nutritional value of these products; however, they typically do not pose serious health risks. In contrast, the presence of harmful chemicals such as urea, formalin, detergents, hydrogen peroxide, acids, and melamine significantly heightens health risks, as these substances can lead to both acute and chronic toxic effects [4]. These harmful chemicals are usually added for economically motivated gain. Additionally, some dietary supplements used by athletes of all ages may contain undeclared or adulterated substances, which can result in negative health effects in both the short and long term [5]. Studies indicate that up to 50% of dietary supplements may be contaminated with melamine, raising serious concerns about consumer safety [6,7].

The use of melamine as a food additive is not approved. However, due to its high nitrogen content, it is sometimes illegally added to protein products to enhance their perceived nutritional value. This practice is employed in the adulteration of foods to increase the total nitrogen content and to mislead analytical methods, such as the Kjeldahl method, which estimates protein content based on total nitrogen levels [8]. Melamine contamination in food primarily results from the use of melamine-contaminated raw materials, particularly milk and dairy products, as well as other protein sources like wheat gluten [9].

Melamine is a small, polar molecule that consists of a triazine ring with three amino groups and exhibits a pronounced ability to form hydrogen bonds. It is a nitrogen-rich compound, with nitrogen comprising approximately 66.67% of its molecular weight [3]. Melamine forms stable crystal structures, particularly in combination with molecules such as cyanuric acid, uracil, riboflavin, barbituric acid, and uric acid [3]. Due to its chemical properties and its capacity to form high-molecular-weight complexes, melamine has become one of the most widely used organic compounds in industrial production, especially in the manufacture of adhesives, various cleaning products, plastics, and fertilizers [10] and in the resin industry [11]. Together with cyanuric acid, it is among the most widely utilized chemicals, with an annual production exceeding 45,000 metric tons [12].

Melamine is widely used in packaging through the application of melamine resins [13,14,15] as an additive in the manufacture of cardboard and paper [16], and as a component in adhesives and coatings utilized in food contact materials [17]. Therefore, there is a possibility that melamine may migrate from food contact materials into food and subsequently be absorbed by the human body [13]. Additionally, melamine can enter food by migrating from kitchenware when exposed to high temperatures, alcohol, and acids. It may also be present as a residue from the use of the pesticide and herbicide cyromazine, which is employed in veterinary medicine as an ectoparasiticide. Furthermore, melamine can contaminate food through the use of disinfectants containing trichloromelamine and dichloroisocyanurate, which can decompose into melamine when used for disinfecting packaging and equipment in the food industry. Contamination of drinking water and water used in the food industry with cyanuric acid can also occur as a result of using the disinfectant sodium dichloroisocyanurate [18].

Melamine exhibits low acute toxicity and is not classified as carcinogenic; however, it can be harmful if ingested, inhaled, or if it comes into contact with the skin, especially in the presence of cyanuric acid [4]. It is not metabolized and has a short half-life. Consequently, when combined with cyanuric acid in humans, it can lead to the formation of insoluble crystals, kidney or bladder stones, and urinary tract obstructions [19]. Additionally, melamine can cause significant damage to kidney cells and the tubular structures within the kidneys [4,20]. Contamination with melamine garnered significant attention in 2008 when elevated levels of this chemical were detected in baby food, milk, and milk powder. By the end of November 2008, there were 294,000 reported cases of urinary stones in infants and young children in China. While most cases were asymptomatic, a small number of children developed acute renal failure, resulting in over 50,000 hospitalizations and six confirmed deaths [9].

Maximum residue limits (MRL) for melamine have been established globally. In China and the United States of America (USA), the MRL for infant formula is set at 1.0 mg/kg, while for milk and other dairy products, it is 2.5 mg/kg [21]. The European Union (EU) has established a maximum limit (ML) of 2.5 mg/kg for all products, with specific limits of 1.0 mg/kg for infant formula and follow-on formula, and 0.15 mg/kg for baby food, depending on whether the product is marketed in powder or liquid form [22]. In light of these risks and instances of milk adulteration, the World Health Organization (WHO) has set a Tolerable Daily Intake (TDI) of 0.2 mg/kg of body weight for melamine, while the USA has established a limit of 0.63 mg/kg of body weight per day [10]. The European Food Safety Authority has determined a maximum daily melamine contamination level of 0.50 mg/kg of body weight per day, below which no adverse health effects are expected [23].

To effectively control and monitor the concentration of melamine residues in milk and dairy products, it is essential to employ sensitive, efficient, and reliable methods for quantification. Techniques for the quantitative determination of melamine include enzyme immunoassay (EIA), infrared (IR) analysis, gas chromatography-mass spectrometry (GC-MS), liquid chromatography-mass spectrometry (LC-MS), and high-performance liquid chromatography (HPLC) with ultraviolet (UV) detection [8,11,19,24,25,26]. Chromatographic methods are widely utilized in food analysis, with liquid chromatography (LC) often preferred over gas chromatography (GC) because it does not require prior derivatization of samples. Numerous LC methods have been developed and implemented for the detection and quantification of melamine in food [27,28]. LC-MS/MS is established as a robust and reliable technique for determining melamine and cyanuric acid because it provides the highest sensitivity, specificity, and accuracy; enables rapid analysis without derivatization; and ensures reliable detection in complex sample matrices [29,30].

The aim of this study is to investigate the occurrence of melamine and cyanuric acid in milk, baby food, and protein supplements collected in Croatia using a straightforward, rapid, accurate, and reliable analytical methodology. This research will be the first to investigate the presence of melamine and cyanuric acid in milk powder, baby food, and protein supplements in Croatia.

## 2. Materials and Methods

### 2.1. Sample Collection

A total of 56 samples (15 milk samples, 15 baby food samples, and 26 protein supplements) were collected from retail chains in the city of Zagreb, Croatia, during the end of 2022 and the beginning of 2023. Collected milk samples were pasteurized cow milk originating from Croatia. Protein supplements were whey-based, while baby food samples were milk-based infant formula placed on the market as powder. Both group samples originated from the EU. All samples were well homogenized. The milk samples were stored in a freezer at −18 °C, while the baby food and protein supplement samples were stored in a desiccator until analysis.

### 2.2. Chemicals and Reagents

The standards melamine, 99%, and cyanuric acid, 98%, as well as the internal standards melamine-^13^C_3_, isotopic purity 99 atom % ^13^C, in solution at a concentration of 1000 μg/mL and cyanuric acid-^13^C_3_, isotopic purity 99 atom % ^13^C, in solution at a concentration of 1000 μg/mL were purchased from Sigma-Aldrich (Steinheim, Germany). Acetonitrile (ACN), HPLC-grade, was supplied by Chromasolv (Honeywell Specialty Chemicals, Seelze, Germany). Methanol (MeOH), ULC/MS-CC/SFC purity, was supplied by Biosolve Chimie (Dieuze, France). Ammonium acetate, ~98%, was supplied by Sigma-Aldrich (Steinheim, Germany). Formic acid, 98–100%, for analysis EMSURE ACS, Reag. Ph Eur, was supplied by Merck (Darmstadt, Germany). Ultrapure water obtained from Milli-Q system (Millipore, Bedford, MA, USA) and nitrogen (N_2_) (purity 5.5) were supplied by SOL S.p.A. (Monza, Italy).

### 2.3. Equipment

Samples were prepared using a Waring Commercial Blender 7011HS (Waring Commercial, Stamford, CT, USA), an IKA vortex model MS2 Minishaker (IKA-WERKE GMBH & CO.KG, Staufen, Germany), a VWR multi-tube vortex (VWR International GmbH, Ulm, Germany), a ROTONTA 460 R centrifuge (Hettich Zentrifugen, Tuttlingen, Germany), and a centrifuge Thermo Fischer Scientific SL16R (Thermo Fisher Scientific, Waltham, MA, USA).

### 2.4. Standard Solutions and Calibration Curve

Individual stock standard solutions of melamine and cyanuric acid were prepared at a concentration of 1000 µg/mL while melamine-^13^C_3_ and cyanuric acid-^13^C_3_ were prepared at a concentration of 20 µg/mL by dissolving in ultrapure water and stored in a refrigerator at a temperature of 4 °C for 1 year.

The individual stock standard solutions were used to prepare working standard solutions for melamine and cyanuric acid at a concentration of 20 µg/mL and 0.2 µg/mL and a mixture of working internal standard solutions for melamine-^13^C_3_ and cyanuric acid-^13^C_3_ of 2 µg/mL. All standard solutions were prepared in 70:30 (*v*/*v*) acetonitrile:water and stored in a refrigerator at 4 °C for 1 month.

A standard calibration curve was prepared at seven concentration levels: 0, 0.001, 0.005, 0.025, 0.05, 0.1, and 0.5 µg/mL, dissolved in 70:30 (*v*/*v*) acetonitrile:water. Linear regression was used to create calibration curves. For melamine, the calibration curve followed the equation Y = 1.404697X + 0.057523, demonstrating excellent linearity with a determination coefficient (R^2^) value of 0.999992. For cyanuric acid, the calibration curve was described by Y = 2.115086X + 0.027402, with a determination coefficient (R^2^) value of 0.999956.

### 2.5. Sample Preparation and Extraction

Milk, baby food and protein supplement samples were homogenized and weighed (1 ± 0.01 g) in a 50 mL polypropylene tube. Samples were spiked with 250 µL of the stock solution of internal standard melamine-^13^C_3_ and 250 µL of the stock solution of internal standard cyanuric acid-^13^C_3_ (concentrations of 20 µg/mL) to achieve the final concentration of 0.1 µg/mL in the sample extract. Samples were incubated for at least 1 h at room temperature. Then, 5 mL of water was added and shaken for 30 s, followed by the addition of 5 mL of acetonitrile and shaken again. The sample was then mixed with 30 mL acetonitrile and 10 mL of hot water (96–100 °C), shaken well for 5 min, and centrifuged at 3400 rpm for 10 min. After the first centrifugation, 1.5 mL of supernatant was transferred to a standard microtube and centrifuged again at 7800 rpm for 10 min; then 1000 µL was transferred from the upper layer to an LC vial.

### 2.6. Quality Control

Sample batches included a calibration blank and a procedural blank to control any contamination during the analytical process. Milk powder containing melamine and cyanuric acid from FAPAS (Fera Science, Sand Hutton, York, UK) was used for quality control. Two reference materials were prepared for each sample batch. Reference values and associated expanded uncertainties for the reference material are 12.4 ± 0.7 mg/kg for melamine and 11.9 ± 0.8 mg/kg for cyanuric acid. All measured values obtained for the reference materials are within these ranges and confirm the adequacy of the applied analytical method. The instrument needle was washed with 50% methanol, 25% acetonitrile, and 25% ultrapure water between every sample.

### 2.7. UHPLC-MS/MS Analysis

After sample preparation, the samples were ready for further analysis. Analysis of melamine and cyanuric acid was performed on a 1290 coupled to a 6460C triple quadrupole LC/MS by Agilent Technologies (Santa Clara, CA, USA), controlled by MassHunter Workstation–LC/MS data acquisition for 6400 series Triple Quadrupole, Ver. 10.1. Chromatographic separation was performed by gradient elution with a flow rate of 0.5 mL min^−1^ on an Acquity UPLC BEH HILIC column (2.1 × 150 mm, 1.7 μm) by Waters (Milford, MA, USA); runtime was 5 min with the a posttime of 2.5 min for column re-equilibration, and the column temperature was 40 °C. The mobile phase (A) consisted of 3% formic acid in ultrapure water, and the mobile phase (B) consisted of 20 mmol/L ammonium acetate in 97% acetonitrile. The gradients were 100% B (0 to 1 min), 5% B and 95% A (1 to 3 min), 5% B and 95% A (3 to 4 min), 100% B (4 to 4.10 min), and 100% B (4.10 to 5 min). The injection volume of the sample extract was 5 µL.

Source parameters were gas temperature 300 °C, gas flow 13 L/min, nebulizer 40 psi, sheath gas temperature 350 °C, sheath gas flow 7.4 L/min, and capillary 2000 V in both positive and negative modes, except for nozzle voltage/charging 0 V (positive) and 1000 V (negative). Delta EMV was set to 300 V in both modes. The optimized parameters for the analytes are listed in Table 1. The chromatogram of cyanuric acid and melamine in the standard solution is shown in Figure 1. An overview of the transitions for melamine and cyanuric acid, along with their internal standards, melamine-^13^C_3_ and cyanuric acid-^13^C_3_, is presented in Figure 2.

### 2.8. Statistical Analysis

Concentrations of melamine and cyanuric acid were calculated as the mean ± standard deviation (SD), median, along with the minimum and maximum values. Statistical analyses were conducted by Stata version 13.1 for Windows (64-bit x86-64) (StataCorp LP, College Station, TX, USA).

## 3. Results and Discussion

A total of 56 samples were analyzed, divided into three groups: 15 samples of milk, 15 samples of baby food, and 26 samples of protein supplements. All samples were analyzed using a quantitative method (UHPLC-MS/MS) to detect the concentrations of melamine and cyanuric acid. For melamine and cyanuric acid, the limit of detection (LOD) and the limit of quantification (LOQ) are 0.06 mg/kg and 0.2 mg/kg, respectively. Descriptive statistics parameters such as range, mean, standard deviation, median, and detection frequency for all three sample groups, analyzed for melamine and cyanuric acid, are summarized in Table 2. All detected concentrations were significantly below the maximum limit (ML) for melamine established by the European Union, which is set at 2.5 mg/kg for all products and 1.0 mg/kg for baby food [22].

Out of a total of 15 milk samples, no melamine concentration above the LOQ was detected in any sample (all results are <0.2 mg/kg). Concentrations of cyanuric acid above the LOQ were detected in five milk samples, which is a detection frequency of 33.3%. The mean value of the detected cyanuric acid concentrations was 0.31 mg/kg with a result range from 0.26 to 0.39 mg/kg. In all 15 baby food samples, no concentration of melamine and cyanuric acid above the LOQ was detected.

Within the 26 samples of protein supplements for six were detected melamine concentrations above the LOQ, which is a detection frequency of 23%. The mean value of the detected melamine concentrations was 0.30 mg/kg with a result within the range of 0.20–0.57 mg/kg. All concentrations of cyanuric acid were below the LOQ (<0.2 mg/kg).

In order to reliably quantify melamine and cyanuric acid at low concentration levels in complex matrices like dairy products and protein-rich supplements, where less selective methods (such as HPLC or UV spectrophotometry) may be affected by matrix effects, UHPLC-MS/MS was important in this study [29,30]. LC-MS/MS has limitations despite its high sensitivity and selectivity, such as its high cost, requirement for highly skilled workers, sensitivity to contamination and matrix effects, and general operational complexity [31]. The study also has certain limitations: other melamine analogs (ammeline, ammelide) were not examined, and the sample size within each product category was limited.

In the last two decades, melamine residues have been identified in milk and dairy products from various countries. Since the 2007–2008 melamine scandal, infant formula and dairy products have been under heightened regulatory scrutiny [32]. Melamine was detected in 9 out of a total of 40 milk powder samples purchased in Uruguay during 2009 [33]. The concentrations ranged from 0.017 to 0.082 mg/kg, while the mean value of 0.028 mg/kg. It was concluded that the consumption of milk powder containing these levels of melamine does not pose a health risk to consumers [34]. In Turkey, 300 samples of milk and dairy products purchased in 2010 from major retailers in Ankara were analyzed [34]. Melamine was not detected in infant formulas or pasteurized UHT milk; however, 8% of the milk powder samples contained melamine, with a mean concentration of 0.694 mg/kg. These findings were the first confirmation of melamine occurrence in milk and dairy products in Turkey. The results are consistent with our study, as no melamine levels were detected in milk and infant formulas.

The study conducted in 2012 demonstrated that nitrogen-rich melamine was added to raw milk to artificially enhance its protein content [28]. Samples collected from China and Saudi Arabia revealed high levels of melamine in infant milk formula, growing-up milk formula, and sweetened full cream milk powder, particularly from China [28]. The mean melamine concentrations, along with their minimum and maximum values, for infant milk formula, growing-up milk formula, and full cream milk powder were 146.93 mg/kg (9.49–258 mg/kg), 96.105 mg kg^−1^ (7.75–251 mg/kg), and 34.18 mg/kg (29.1–39.7 mg/kg), respectively. The results indicate that melamine was detected at levels exceeding 1 mg/kg, surpassing the control limit established in China for powdered infant formula (1 mg/kg) and for other foods (2.5 mg/kg).

In 2015, a total of 138 food supplement samples were analyzed in a study conducted in the Republic of South Africa, revealing that melamine was detected in 64 samples (47%) [5]. The mean concentration of melamine was found to be 6.031 mg/kg. These levels remained within the acceptable limits of daily intake (TDI) of 0.2 mg/kg of body mass, as defined by the WHO. In the same year, a study conducted in India analyzed 10 marketed milk powder samples. The melamine concentrations, determined by HPLC, were significantly lower than the maximum permissible limit of 2.5 mg/kg established by the Food Safety and Standards Authority of India, with results ranging from 0.0001 to 0.0006 mg/kg [35].

A study conducted in Albany, New York, USA, in 2018 analyzed 121 samples categorized into six groups, collected from several major grocery stores. The results indicated that more than half of the human exposure to melamine and cyanuric acid originated from these food products [32]. Specifically, cyanuric acid and melamine accounted for 51% and 26% of the total concentration, respectively, which includes melamine and its three derivatives: ammelin, ammelide, and cyanuric acid. In infant formulas and dairy products, the median concentrations of melamine were reported at 2.70 µg/kg fat weight and 6.28 µg/kg fat weight, respectively. The total melamine concentrations detected in infant formula and dairy products were lower than those found in other food categories analyzed in this study.

In a study from 2018, 69 samples of infant and follow-on formula available on the Iranian market were analyzed [25]. Melamine was analyzed by the HPLC method and was detected in 65% of the samples, with 14.5% of the results exceeding the maximum allowed value of 1 mg/kg set by EU [22]. The average concentration was 0.73 mg/kg, while the maximum measured concentration was 3.63 mg/kg. It was concluded that lower concentrations of melamine reached the analyzed samples by migration from sources such as plastic food packaging or by degradation of some chemicals, and that such low concentrations of melamine do not represent a health risk for consumers [25].

The analysis of melamine and cyanuric acid was conducted in 2022 at the Regional Center for Food in Giza, Egypt, involving 140 powdered dairy products [36]. Melamine was detected in 30% of whole milk powder samples, 73.3% of skim milk powder samples, 66.7% of powdered infant formula samples, and 83.3% of milk-cereal-based infant formula samples. All positive samples contained melamine at a concentration of 0.25 mg/kg, which exceeded the LOD of the LC-MS/MS method employed. No samples exhibited detectable levels of cyanuric acid (<0.05 mg/kg). However, the specific concentrations of melamine in the positive samples were not reported in the study. The study concluded that enhanced regulatory measures are essential at both national and international levels to prevent the use of melamine in animal feeds, fertilizers, and pesticides. It also underscored the importance of routine monitoring of dried milk and infant formulas using rapid and sensitive analytical techniques [36].

In a more recent study conducted in 2024, 40 samples of infant formula and milk powder from various regions of Iran were analyzed [37]. The melamine content in the infant formula samples ranged from 0.001 to 0.004 mg/kg, while the milk powder samples exhibited melamine levels ranging from 0.001 to 0.095 mg/kg. Furthermore, melamine had been detected in 80% of the milk powder samples and in 65% of the infant formula samples, although these levels were below the maximum limits established by the EU [22]. This finding suggests that the consumption of these dairy products does not pose a health risk to consumers, consistent with the results of our study.

The levels of melamine and cyanuric acid observed in this study offer information about contamination patterns across different food matrices. Melamine in milk and baby food was below the LOQ (0.2 mg/kg), consistent with studies reporting non-detectable or very low levels [33,34,35,37], including infant formula in Turkey and India, where melamine was either entirely absent or present at concentrations substantially below national and international regulatory limits [34,35]. In contrast, studies from China, Saudi Arabia, and Iran documented sporadic exceedances of regulatory limits (1 mg/kg) in infant formula [25,28], indicating that contamination patterns remain regionally heterogeneous. In protein supplements, melamine was detected in 23% of samples (0.20–0.57 mg/kg), considerably lower than reported in South Africa (>6 mg/kg) [5], suggesting contamination likely arises from trace background sources rather than deliberate adulteration.

Due to the incidents involving melamine in pet food in 2004 and 2007 [38], as well as the detection of high melamine concentrations in baby food, milk, and milk powder in 2008 [9], concerns have arisen regarding the potential migration of melamine from packaging into food [16]. Considering the widespread use of melamine in food contact materials—including melamine resins in packaging, cardboard, paper, adhesives, and coatings [13,17]—migration represents a plausible contamination pathway. Melamine may leach into food under conditions involving elevated temperatures, acidic media, or alcohol. Furthermore, residues from the pesticide cyromazine or from disinfectants containing trichloromelamine and dichloroisocyanurate may contribute to contamination. Cyanuric acid was detected in 33.3% of milk samples and below the LOQ (0.2 mg/kg) in other products. Its presence in milk samples suggests that decomposition of disinfectants such as sodium dichloroisocyanurate or sodium hypochlorite could represent an alternative contamination source [18]. Therefore, both material-related and sanitation-related pathways should be considered when evaluating potential contamination. Additionally, the documented negative effects of melamine and cyanuric acid on human health [19,20] underscore the necessity for monitoring and control. Moving forward, greater attention should be directed toward protein supplements, given their rising consumption [5] and the risk of melamine adulteration intended to falsely enhance protein content [6]. This concern justifies their inclusion in regular food safety monitoring protocols.

## 4. Conclusions

In this study, a quantitative UHPLC-MS/MS method was employed to analyze melamine and cyanuric acid in milk, baby food, and protein supplements during 2022 and 2023. Only protein supplements exhibited melamine concentrations above the LOQ, while no melamine concentrations were detected in the milk and baby food samples. Cyanuric acid was identified in 33.3% of the milk samples but remained below quantifiable levels in the other product categories. These findings, which align with studies conducted in the USA, Iran, Uruguay, Turkey, and South Africa, indicate that melamine continues to be a global concern. However, the concentrations detected in the majority of food products today do not pose an immediate risk to consumer health. Nonetheless, the frequent detection of melamine, even at low levels, underscores the necessity for ongoing monitoring, enhanced enforcement of regulations, and further research into potential sources of contamination, such as packaging or processing methods. Regular monitoring using sensitive and validated analytical techniques remains essential to ensure food safety, particularly for vulnerable populations such as infants, due to the cumulative exposure risk and the potential for intentional adulteration.

The conclusions of the study, supported by data from various international reports, indicate that melamine contamination in milk and infant formula products persists in many regions worldwide. However, these contamination levels are generally below the maximum residue limits established by regulatory authorities, including the Codex Alimentarius Commission and the European Food Safety Authority.

## Figures and Tables

**Figure 1 foods-14-04292-f001:**
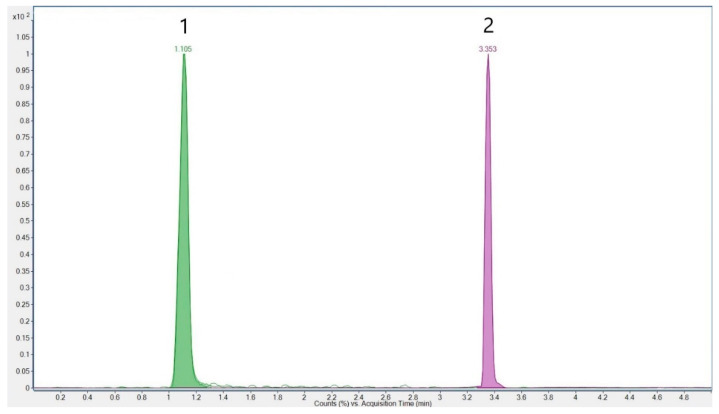
Chromatogram of cyanuric acid (1) and melamine (2)–standard solution concentration of 0.05 µg/mL.

**Figure 2 foods-14-04292-f002:**
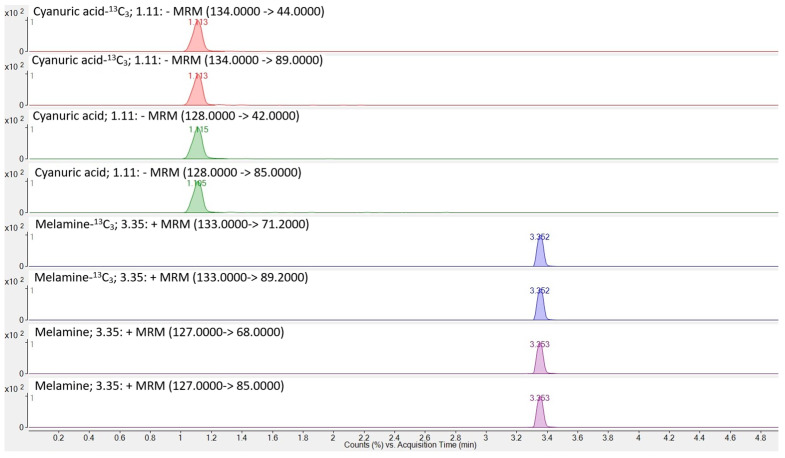
Overview of transitions for the analytes melamine and cyanuric acid and their internal standards melamine-^13^C_3_ and cyanuric acid-^13^C_3_.

**Table 1 foods-14-04292-t001:** Optimized MS/MS parameters for melamine and cyanuric acid and internal standards melamine-^13^C_3_ and cyanuric acid-^13^C_3_.

Analyte	Precursor Ion (*m/z*)	Product Ion (*m/z*)	Fragmentor (V)	Collision Energy (eV)	Collision Cell Accelerator Voltage (V)	Polarity	Retention Time (min)
Melamine	127	**85**	100	20	7	Positive	3.35
	127	68	100	30	7	Positive	3.35
Cyanuric acid	128	85	70	6	7	Negative	1.11
	128	**42**	70	10	7	Negative	1.11
Melamine-^13^C_3_	133	**89.2**	120	20	7	Positive	3.35
	133	71.2	120	34	7	Positive	3.35
Cyanuric acid-^13^C_3_	134	89	80	8	7	Negative	1.11
	134	**44**	80	35	7	Negative	1.11

Note: Product ion (bolded) was used for compound quantification.

**Table 2 foods-14-04292-t002:** Overview of the results obtained for melamine and cyanuric acid by product group.

Product Group	Parameters	Melamine	Cyanuric Acid
Milk-pasteurized cow milk (*n* = 15)	Range (mg/kg)	n.d.	0.26–0.39
	Mean ± SD (mg/kg)	n.d.	0.31 ± 0.056
	Median (mg/kg)	n.d.	0.28
	df (%)	0	33.3
	LOD (mg/kg)	0.06	0.06
	LOQ (mg/kg)	0.2	0.2
Baby food—milk-based infant formula placed on the market as powder (*n* = 15)	Range (mg/kg)	n.d.	n.d.
	Mean ± SD (mg/kg)	n.d.	n.d.
	Median (mg/kg)	n.d.	n.d.
	df (%)	0	0
	LOD (mg/kg)	0.06	0.06
	LOQ (mg/kg)	0.2	0.2
Protein supplements—whey-based (*n* = 26)	Range (mg/kg)	0.20–0.57	n.d.
	Mean ± SD (mg/kg)	0.30 ± 0.14	n.d.
	Median (mg/kg)	0.26	n.d.
	df (%)	23.0	0
	LOD (mg/kg)	0.06	0.06
	LOQ (mg/kg)	0.2	0.2

df—detection frequencies; n.d.—not detected (results below the LOQ).

## Data Availability

The original contributions presented in the study are included in the article, further inquiries can be directed to the corresponding author.

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
