# Peer review of "Occurrence of Melamine and Cyanuric Acid in Milk, Baby Food, and Protein Supplements Marketed in Croatia"

_foods, 2025, doi:10.3390/foods14244292_

Round 1
Reviewer 1 Report
Comments and Suggestions for Authors
This manuscript presents a well-designed and timely study on the occurrence of melamine (MEL) and cyanuric acid (CYA) in milk, baby food, and protein supplements from the Croatian market. The topic is highly relevant to the scope of Foods, particularly in the areas of food safety, chemical contaminants, and food adulteration.
The study provides the first surveillance data for these contaminants in this specific market, which represents a valuable and novel contribution to the field. The rationale for the study is clearly articulated, grounding the work in the context of both historical adulteration scandals and ongoing concerns about low-level contamination from sources like packaging migration.
The methodology is robust, the results are presented clearly, and the discussion is thorough. The manuscript is well-written, and the conclusions are well-supported by the data. It is suitable for publication after addressing a few minor points.
Minor Comments & Specific Corrections
The manuscript is in very good condition, but a few minor corrections are required to improve clarity and professionalism.
- Typographical Error: Line 254: "...with a result within te range of 0.20-0.57 mg/kg. Please change "te" to "the".
- Grammatical Error: Line 329: "...although these levels were below the maximum limits establish by EU. This should be in the past participle. Please change to "established by the EU".
- Awkward Phrasing (Clarity): Line 304: "In a study from 2018, 69 infants along with follow up formula samples available on the Iranian market were analyzed. This phrasing is awkward and could be misinterpreted as the infants themselves being analyzed. Please rephrase for clarity, for example: "69 samples of infant and follow-on formula available on the Iranian market were analyzed..."
- Contradictory Statement in Conclusion: Line 345-346: "Only protein supplements exhibited melamine concentrations above the LOQ, while no concentrations were detected in the milk and baby food samples." This statement contradicts the Abstract, which states: "Cyanuric acid concentrations above the limit of quantification (LOQ) were found in five milk samples...". The conclusion must be precise. Please revise the sentence to specify melamine, e.g.: "...while no melamine concentrations were detected in the milk and baby food samples."
- Run-on Sentence (Style): Line 133-135: "All samples were well homogenized, the milk samples were stored in a freezer at -18 C while the baby food and protein supplement samples were stored in a desiccator until analysis." This sentence combines multiple actions. For better readability, please split it: "All samples were well homogenized. The milk samples were stored in a freezer at -18 C, while the baby food..."
I mentioned the language corrections above.
Author Response
Reviwer 1:
Changes in article are marked in red.
- Typographical Error: Line 254: "...with a result within te range of 0.20-0.57 mg/kg. Please change "te" to "the".
- Line 266: corrected to: „the“
- Grammatical Error: Line 329: "...although these levels were below the maximum limits establish by EU. This should be in the past participle. Please change to "established by the EU".
- Line 347: changed to: „Furthermore, melamine had been detected in 80% of the milk powder samples and in 65% of the infant formula samples, although these levels were below the maximum limits established by the EU.“
- Awkward Phrasing (Clarity): Line 304: "In a study from 2018, 69 infants along with follow up formula samples available on the Iranian market were analyzed. This phrasing is awkward and could be misinterpreted as the infants themselves being analyzed. Please rephrase for clarity, for example: "69 samples of infant and follow-on formula available on the Iranian market were analyzed..."
- Line 324: rephrased to: „69 samples of infant and follow-on formula“
- Contradictory Statement in Conclusion: Line 345-346: "Only protein supplements exhibited melamine concentrations above the LOQ, while no concentrations were detected in the milk and baby food samples." This statement contradicts the Abstract, which states: "Cyanuric acid concentrations above the limit of quantification (LOQ) were found in five milk samples...". The conclusion must be precise. Please revise the sentence to specify melamine, e.g.: "...while no melamine concentrations were detected in the milk and baby food samples."
- Line 389: specified to refer exactly to melamine to make it clearer and more precise: „…while no melamine concentrations…“
- Run-on Sentence (Style): Line 133-135: "All samples were well homogenized, the milk samples were stored in a freezer at -18 C while the baby food and protein supplement samples were stored in a desiccator until analysis." This sentence combines multiple actions. For better readability, please split it: "All samples were well homogenized. The milk samples were stored in a freezer at -18 C, while the baby food..."
- Line 139-141: separated the sentences for better readability: „All samples were well homogenized. The milk samples were stored in a freezer at –18 °C, while the baby food and protein supplement samples were stored in a desiccator until analysis.“
Reviewer 2 Report
Comments and Suggestions for Authors
The authors used UHPLC-MS/MS to investigate the contamination of melamine and cyanuric acid in commercially available milk, infant formula, and protein supplements in Croatia. Overall, there are no major problems with the article. However, some small details need to be revised.
The specific content is as follows:
- It is recommended to provide more detailed information about the sample brand (e.g., local or international brand?) or type (e.g., is the milk UHT or pasteurized? Is the protein supplement whey, plant-based, or a combination?) in the “Sample Collection” section (2.1). This will help readers better understand the representativeness of the results and may provide clues about the source of contamination. Please specify the exact type of "baby food" sample (e.g., cereal-based, fruit puree, or meat puree?), as EU regulations may have slightly different limits for different types of baby food.
- The article mentions that "the runtime was 5 minutes with a posttime of 2.5 minutes".Is the post-processing time the time for balancing the instrument under initial conditions?
- The authors prepared standard working curves for seven concentration points. Please provide the equation of the standard curve and thedetermination coefficient (R2) in the results.
- Although the possibility of migration from packaging is briefly mentioned on page 10, this discussion could be expanded upon slightly. Given that cyanuric acid, rather than melamine, was detected in the milk, its possible sources (e.g., from the decomposition of the disinfectant sodium hypochlorous acid) can be discussed speculatively, echoing the content of the introduction (page 3).
- Please carefully check the reference format according to the journal.
Author Response
Rewiver 2
Changes in article are marked in red.
The specific content is as follows:
- It is recommended to provide more detailed information about the sample brand (e.g., local or international brand?) or type (e.g., is the milk UHT or pasteurized? Is the protein supplement whey, plant-based, or a combination?) in the “Sample Collection” section (2.1). This will help readers better understand the representativeness of the results and may provide clues about the source of contamination. Please specify the exact type of "baby food" sample (e.g., cereal-based, fruit puree, or meat puree?), as EU regulations may have slightly different limits for different types of baby food.
- L136-138: added: „Collected milk samples were pasteurized cow milk originating from Croatia. Protein supplements were whey-based, while baby food samples were milk-based infant formula placed on the market as powder. Both group samples originated from the EU.“
- The article mentions that "the runtime was 5 minutes with a posttime of 2.5 minutes".Is the post-processing time the time for balancing the instrument under initial conditions?
- L213-214: added explanation: „runtime was 5 min with the a posttime of 2.5 min for column re-equilibration“
- The authors prepared standard working curves for seven concentration points. Please provide the equation of the standard curve and the determination coefficient (R2) in the results.
- L175-179 : added equation of the standard curve and the determination coefficient (R2) for melamine and cyanuric acid: „ Linear regression was used to create calibration curves. For melamine, the calibration curve followed the equation Y = 1.404697X + 0.057523, demonstrating excellent linearity with a determination coefficient (R2) value of 0.999992. For cyanuric acid, the calibration curve was described by Y = 2.115086X + 0.027402, with a determination coefficient (R2) value of 0.999956.“
- Although the possibility of migration from packaging is briefly mentioned on page 10, this discussion could be expanded upon slightly. Given that cyanuric acid, rather than melamine, was detected in the milk, its possible sources (e.g., from the decomposition of the disinfectant sodium hypochlorous acid) can be discussed speculatively, echoing the content of the introduction (page 3).
- L367-377: added expanded discussion: „Considering the widespread use of melamine in food contact materials—including melamine resins in packaging, cardboard, paper, adhesives, and coatings [13,17]—migration represents a plausible contamination pathway. Melamine may leach into food under conditions involving elevated temperatures, acidic media, or alcohol. Furthermore, residues from the pesticide cyromazine or from disinfectants containing trichloromelamine and dichloroisocyanurate may contribute to contamination. Cyanuric acid was detected in 33.3% of milk samples and below the LOQ (0.2 mg/kg) in other products. Its presence in milk samples suggests that decomposition of disinfectants such as sodium dichloroisocyanurate or sodium hypochlorite could represent an alternative contamination source [18]. Therefore, both material-related and sanitation-related pathways should be considered when evaluating potential contamination.“
- Please carefully check the reference format according to the journal.
- L422-503: references format changes according to the journal and added new references
Reviewer 3 Report
Comments and Suggestions for Authors
- The manuscript needs to clarify the aim of the study. Is this to investigate the occurrence of melamine and cyanuric acid in products in Croatia (L14-15), or to develop a methodology for quantifying melamine and cyanuric acid in milk (L124-125).
- The introduction flow can be improved. Currently it wrote too much to introduce use of melamine and how they can be introduced into foods. These can be discussed in one paragraph. The methodology of using LC/MS-MS need to be justified in Introduction or Discussion. The rationale for the determination of cyanuric acid needs to be provided.
- L213-214: Which ion pairs are used for quantification?
- Discussion: Discussion can focus on levels in different products and samples, method precision and accuracy, differences in species of the contaminants across different samples, why LC-MS/MS is needed, the limitations of the study, etc. The levels in other studies can be discussed in brief for comparison purposes.
- Table 2. More details can be provided, since there are only 56 samples.
- Minor issue: L53-L55: These harmful chemicals are usually added for economically motivated gains.
- Minor issue: L47-48: Provide reference to back up this claim.
Author Response
Reiwer 3.
Changes in article are marked in red.
- The manuscript needs to clarify the aim of the study. Is this to investigate the occurrence of melamine and cyanuric acid in products in Croatia (L14-15), or to develop a methodology for quantifying melamine and cyanuric acid in milk (L124-125).
- L127-131: Modified text to clarify the aim of the study, which focuses on investigationg the occurrence of melamine and cyanuric aci din milk, baby food and protein supplements: „The aim of this study is to investigate the occurrence of melamine and cyanuric acid in milk, baby food, and protein supplements collected in Croatia using a straightforward, rapid, accurate, and reliable analytical methodology. This research will be the first to investigate the presence of melamine and cyanuric acid in milk powder, baby food, and protein supplements in Croatia.“
- The introduction flow can be improved. Currently it wrote too much to introduce use of melamine and how they can be introduced into foods. These can be discussed in one paragraph. The methodology of using LC/MS-MS need to be justified in Introduction or Discussion. The rationale for the determination of cyanuric acid needs to be provided.
- L123-126: added why are we using LC/MS-MS method: „LC-MS/MS is established as a robust and reliable technique for determining melamine and cyanuric acid because it provides the highest sensitivity, specificity, and accuracy; enables rapid analysis without derivatization; and ensures reliable detection in complex sample matrices.“
- Explanation: Although cyanuric acid does not have an ML, it can also, like melamine, enter food as described in L87-89, and when combined with melamine, it can lead to the formation of insoluble crystals, kidney or bladder stones, and urinary tract obstructions as mentioned in L92-94. It is for these reasons that we decided to include the determination of cyanuric acid in the study. That is why I wrote that much about how melamine and cyanuric acid can be introduced into foods.
- L213-214: Which ion pairs are used for quantification?
- L222-225: In Table 1 were specified ion pairs used for quantification (bolded and noted: „Note: Product ion (bolded) was used for compound quantification.“
- Discussion: Discussion can focus on levels in different products and samples, method precision and accuracy, differences in species of the contaminants across different samples, why LC-MS/MS is needed, the limitations of the study, etc. The levels in other studies can be discussed in brief for comparison purposes.
- L-352-363: added: „The levels of melamine and cyanuric acid observed in this study offer information about contamination patterns across different food matrices. Melamine in milk and baby food was below the LOQ (0.2 mg/kg), consistent with studies reporting non-detectable or very low levels [32,33,35,37], including infant formula in Turkey and India, where melamine was either entirely absent or present at concentrations substantially below national and international regulatory limits [33,35]. In contrast, studies from China, Saudi Arabia, and Iran documented sporadic exceedances of regulatory limits (1 mg/kg) in infant formula [25,28], indicating that contamination patterns remain regionally heterogeneous. In protein supplements, melamine was detected in 23% of samples (0.20–0.57 mg/kg), considerably lower than reported in South Africa (>6 mg/kg) [34], suggesting contamination likely arises from trace background sources rather than deliberate adulteration. „
- L-373-377: added: „Cyanuric acid was detected in 33.3% of milk samples and below the LOQ (0.2 mg/kg) in other products. Its presence in milk samples suggests that decomposition of disinfectants such as sodium dichloroisocyanurate or sodium hypochlorite could represent an alternative contamination source [18]. Therefore, both material-related and sanitation-related pathways should be considered when evaluating potential contamination.“
- L-273-281: added: „In order to reliably quantify melamine and cyanuric acid at low concentration levels in complex matrices like dairy products and protein-rich supplements, where less selective methods (such as HPLC or UV spectrophotometry) may be affected by matrix effects, UHPLC-MS/MS was important in this study [29,30]. LC-MS/MS has limitations despite its high sensitivity and selectivity, such as its high cost, requirement for highly skilled workers, sensitivity to contamination and matrix effects, and general operational complexity [31]. The study also has certain limitations: other melamine analogues (ammeline, ammelide) were not examined, and the sample size within each product category was limited.“
- Table 2. More details can be provided, since there are only 56 samples.
- L266-268: Table 2 is expanded with LOD and LOQ values and description of samples
- Minor issue: L53-L55: These harmful chemicals are usually added for economically motivated gains.
- Line 56: suggested sentence added: “These harmful chemicals are usually added
for economically motivated gains.“
- Minor issue: L47-48: Provide reference to back up this claim.
- L47-49: reference provided: [2,3]
Round 2
Reviewer 3 Report
Comments and Suggestions for Authors
Although the Introduction still contains too much detail about melamine and the Discussion has excessive information regarding melamine levels in different countries, the manuscript has improved and is suitable for publication.